# Line-Field Confocal Optical Coherence Tomography: A New Skin Imaging Technique Reproducing a “Virtual Biopsy” with Evolving Clinical Applications in Dermatology

**DOI:** 10.3390/diagnostics14161821

**Published:** 2024-08-21

**Authors:** Simone Cappilli, Andrea Paradisi, Alessandro Di Stefani, Gerardo Palmisano, Luca Pellegrino, Martina D’Onghia, Costantino Ricci, Linda Tognetti, Anna Elisa Verzì, Pietro Rubegni, Veronique Del Marmol, Francesco Lacarrubba, Elisa Cinotti, Mariano Suppa, Ketty Peris

**Affiliations:** 1Dermatologia, Dipartimento di Medicina e Chirurgia Traslazionale, Università Cattolica del Sacro Cuore, 00168 Rome, Italy; aparad78@gmail.com (A.P.); alessandro.distefani@gmail.com (A.D.S.); palmisanogerardo@gmail.com (G.P.); luca.pellegrino777@gmail.com (L.P.); ketty.peris@unicatt.it (K.P.); 2UOC di Dermatologia, Dipartimento di Scienze Mediche e Chirurgiche, Fondazione Policlinico Universitario A. Gemelli—IRCCS, 00168 Rome, Italy; 3Dermatology Unit, Department of Medical, Surgical and Neurological Sciences, University of Siena, 53100 Siena, Italy; martina.donghia@gmail.com (M.D.); pietro.rubegni@gmail.com (P.R.); elisa.cinotti@unisi.it (E.C.); 4Department of Experimental, Diagnostic and Specialty Medicine (DIMES), S. Orsola-Malpighi University Hospital, University of Bologna, 40126 Bologna, Italy; costanricci@gmail.com; 5Dermatology Clinic, University of Catania, 95131 Catania, Italy; annaeverzi@gmail.com (A.E.V.); franclacarrubba@gmail.com (F.L.); 6Department of Dermatology, Erasme Hospital, Université Libre de Bruxelles, 1050 Brussels, Belgium; v.marmol@skynet.be (V.D.M.); dr.marianosuppa@gmail.com (M.S.); 7Groupe d’Imagerie Cutanée Non Invasive (GICNI), Société Française de Dermatologie (SFD), 42055 Paris, France; 8Department of Dermatology, Institut Jules Bordet, Université Libre de Bruxelles, 1050 Brussels, Belgium

**Keywords:** diagnostic imaging, histology of the skin, line-field confocal optical coherence tomography, clinical dermatology, review, virtual biopsy

## Abstract

Background: Line-field confocal optical coherence tomography is a novel technology able to reproduce a “virtual biopsy” of the skin. The aim of this review is to explore the application of line-field confocal optical coherence tomography (LC-OCT) in various skin diseases, covering skin cancers, inflammatory and infectious skin diseases, genetic diseases, cosmetic procedures, and less common disorders. Methods: Study selection was conducted based on LC-OCT and using pertinent MeSh terms, following Preferred Reporting Items for Systematic Reviews and Meta-Analyses (PRISMA) guidelines from inception to March 2024; to evaluate the quality and risk of bias of studies, Quality Assessment of Diagnostic Accuracy Studies-2 (QUADAS-2) was used. Results: the search retrieved 154 papers according to the selection criteria; after removing publications by one or more of the exclusion criteria, a total of 96 studies were found to be suitable for the analysis. Conclusions: Increasing evidence supports the use of LC-OCT as an adjunctive diagnostic tool for the in vivo diagnosis of a variety of skin tumors. As this device can be considered a “bridge” between dermoscopy and histopathology, widening applications in numerous fields of clinical dermatology, including inflammatory skin disease treatment, presurgical mapping, cosmetic procedures, and monitoring of non-invasive therapies, have been explored.

## 1. Introduction

In recent decades, the development of new technologies has increased the diagnostic capability in dermatology [1,2]. Emerging imaging techniques include digital photographic imaging, reflectance confocal microscopy (RCM), optical coherence tomography (OCT), high frequency ultrasonography, and artificial intelligence (AI) [2,3]. In daily practice, dermatologists rely primarily on clinical and dermoscopic examination, whereas biopsy remains indicated in clinical scenarios when histopathologic information is required to make a diagnosis and establish a plan for a definite treatment [2,3]. However, skin biopsy may have an associated procedural issue resulting from biopsy site selection, technique, or choice of transport media, and is time-consuming [4,5]. Indeed, this diagnostic technique requires a specific preparation of the sample, as thin slices of tissue are cut from the paraffin block, immobilized on slides, stained commonly with hematoxylin and eosin, and consequently sealed with a cover slip [4]. Such technical steps imply a delay between the procedure and the examination result [4,5]. Moreover, clinicians should consider that not all skin biopsies lead to a definite diagnosis, and that dermatologists cannot routinely use this method for every patient they consult [6,7]. In this context, new non-invasive diagnostic tools have contributed to providing useful information for a rapid in vivo diagnosis and for the identification of a biopsy site, which can aid pathology evaluation and save time for confirmatory diagnosis, anticipating treatment and improving patient management [1,2,3,8]. Among these tools, dermatoscopy currently represents the most common and important instrument that clinicians use in the clinical daily practice [9]. Since its introduction, it has gained popularity for the differentiation of melanocytic nevi and melanomas, demonstrating a greater diagnostic accuracy than the unaided eye [10]. Subsequently, the indications for its use have broadened to include the evaluation of hair disorders, nail pigmentation, disorders of appendages, inflammatory dermatoses, and collagen vascular diseases, and also for the assessment of therapy response [2,3,9,10,11]. Such progress has modified the approach from clinicopathological correlation towards a clinical-dermatoscopic-pathological correlation [2,9,10,11]. 

RCM is an established real-time imaging technique with a perspective parallel to the skin surface. Being an 830 nm laser, images are created with high resolution of the skin and cutaneous structures (lateral resolution 0.5–1 µm, axial resolution 3–4 µm), thanks to differences between refractive indices of cellular structures [2,3,11]. RCM imaging has been shown to significantly improve diagnostic accuracy and early detection of melanocytic and nonmelanocytic skin cancers compared with clinical and dermatoscopy alone [12,13]. The simultaneous application of dermatoscopy and RCM improves the ability of clinicians to differentiate benign from malignant skin lesions, reducing the number of unnecessary biopsies by 50% to 70% [12,13]. This technique is also largely applied for the diagnosis of inflammatory and infectious skin disorders, including eczema, psoriasis, and scabies, as well as for the assessment of mucosal lesions in oral or genital areas [11,13]. Compared to RCM, OCT allows for real-time, in vivo, vertical imaging of skin morphology approaching that of histopathology. It has increased imaging depth (2 mm) compared to RCM, at the expense of decreased cellular resolution (7.5 μm), even though new commercial systems may offer higher definition (2 µm) [14,15]. OCT has been extensively investigated in the diagnosis of epithelial carcinomas, particularly basal cell carcinoma (BCC), while its application remains unsatisfactory for a robust diagnosis of malignant melanoma [15].

Line-field confocal optical coherence tomography (LC-OCT, DAMAE Medical^®^, Paris, France) is a novel technology developed in recent years and approved as a medical device in Europe in 2020 [16,17]. As it integrates the technical properties of RCM in terms of cellular resolution, and of OCT for depth acquisition, LC-OCT is able to reproduce a “virtual biopsy” of the skin [17,18]. Numerous scientific articles have been published on this technique so far, reflecting the growing interest in its application in dermatology [18,19].

The aim of this review is to explore the application of LC-OCT in various skin diseases, covering skin cancers, inflammatory and infectious skin diseases, genetic diseases, cosmetic procedures, and less common disorders.

### 1.1. LC-OCT Technical Properties

**LC-OCT device**. The LC-OCT device incorporates a two-beam interference microscope that integrates the fundamental principles of OCT and line-scanning RCM [16,17]. The confocal configuration operates by illuminating a specific point in the skin tissue, precisely at the point of focus of the microscope objective, and subsequently detecting the light scattered back from this point, while spatially eliminating lights originating outside the focal point. This is typically obtained by placing a pinhole in front of the detector. A detailed report of the technical properties and physical mechanisms is provided elsewhere [16,17].

**LC-OCT image acquisition**. This technique acquires vertical and horizontal images and videos up to the superficial/mid dermis with high resolution, thanks to technical skills merging OCT depth acquisition (≅500 µm) and RCM isotropic resolution (≅1.3 µm lateral resolution, ≅1.1 µm axial resolution). In addition, 3D cubes/slices are generated with dedicated software. On the screen the image is visualized together with the corresponding dermatoscopic image, indicating the position of the probe tip during the exam [16,17]. 

### 1.2. LC-OCT Practical Properties

The LC-OCT optical setup is integrated inside a handheld probe that weighs less than 1.2 kg and can be placed on any part of the body. A drop of paraffin oil must be applied on the skin before imaging to ensure the refractive index matching. A button on the handle of the probe is pressed to start recording videos and to switch between the different imaging modalities (vertical and horizontal). Another button allows the user to adjust the position of acquisition in the vertical mode, or the depth of imaging in the horizontal mode, with a 1 µm precision. Typically, a single exam lasts between 5 and 10 min, and soon after LC-OCT images/videos can be reviewed. For specific needs, various custom image processing algorithms are available to facilitate image analysis [16,17].

## 2. Materials and Methods

### 2.1. Search Methods, Types of Studies, and Participants

This study followed the Preferred Reporting Items for Systematic Reviews and Meta-analyses (PRISMA) guidelines [20]. The search was based on line-field confocal optical coherence tomography and its acronym LC-OCT, using the following Medical Subject Heading (MeSH) terms: (“line-field confocal optical coherence tomography”, “LC-OCT”); (“line-field confocal optical coherence tomography”, “LC-OCT”) AND (“inflammatory” OR “infectious” OR “infestation” OR “melanocytic” OR “non-melanocytic” OR “melanoma” OR “basal cell carcinoma” OR “squamous cell carcinoma” OR “epithelial” OR “skin cancer” OR “vascular lesions” OR “genetic” OR “cosmetic”). A systematic literature analysis was performed in four electronic databases: MEDLINE (PubMed), Embase, Scopus, and Cochrane library, which were searched from inception to 30 March 2024, including all types of publications (original article, case series, and case report) in the English language, that evaluated the application of LC-OCT in the clinical setting. For each article, the reference list was checked to include any possible study ignored by the initial search. Exclusion criteria were animal, in vitro or ex vivo studies, not in English language, or narrative or systemic literature reviews, meta-analysis, or conference abstracts.

### 2.2. Data Collection and Analysis

Study selection was performed in two steps after removing duplicate results in EndNote X9 software (Clarivate Analytics, Philadelphia, PA, USA). The first step involved screening and filtering titles and abstracts (SC and GP). During the second step, two reviewers (LP and ADS) independently assessed the eligibility of full-text manuscripts of the studies identified, registering the reasons for exclusions.

To assess the quality and risk of bias of studies, Quality Assessment of Diagnostic Accuracy Studies-2 (QUADAS-2) was used. This tool comprises 4 domains: patient selection, index test, reference standard, and flow and timing. The risk of bias was scored as unclear, high, or low. Case reports and case series were excluded from quality evaluation.

## 3. Results

We performed a search of articles with the MeSH terms previously mentioned. The search retrieved 154 papers according to the selection criteria; after removing duplicate records, 121 publications remained, which were screened by title and abstract. Additionally, 25 articles were excluded by one or more of the exclusion criteria; finally, a total of 96 studies were found to be suitable for the analysis and were included in this review, as shown in the PRISMA flowchart summarizing the results of this process (Figure 1). Studies included in this review were of high quality, using the QUADAS-2 tool. One study had an unclear risk of bias in patient selection as it was not specified if selection was consecutive or random.

### 3.1. Melanocytic Lesions

LC-OCT findings of dermal and compound nevi were firstly reported in seven melanocytic lesions [21]. A wave-like pattern was identified as the main LC-OCT criterion, referring to alternating hyper- and hypo-reflective lines in the dermis, as corresponding to melanocytic strands/cords/nests in histopathology [21]. This finding was also described in six eyelid compound/dermal nevi along with a regular honeycomb pattern and a well-defined DEJ [22]. A blue nevus showed sheet dermal melanocytes, melanophages, and fibrosis [22]. 

The potential role of this device to distinguish nevi and melanoma was assessed in a study including 84 suspicious melanocytic lesions planned for surgical excision. Criteria indicating a melanoma were an irregular honeycomb pattern, epidermal pagetoid spreading, and absence of dermal nests [23]. Pagetoid spreading also represented the main clue in a single case of lentigo maligna and in a case of eyelid melanoma [22,24]. A nodular melanoma resembling a pigmented BCC was diagnosed for the detection of dermal lobules containing numerous bright pleomorphisms [25]. Correlations between dermoscopy, 3D LC-OCT, and histopathology allowed LC-OCT to be considered a diagnostic bridge in between the two techniques, as it may detect the microscopic subclinical key architectural/cellular changes in benign and malignant melanocytic lesions [26,27] (Table 1). 

Applied to pigmented lesions of the genital area (seven benign melanotic macules, one genital nevus, one invasive melanoma), LC-OCT proved to be a painless valuable tool for evaluating challenging skin lesions of this sensitive zone [28] (Figure 2).

### 3.2. Non-Melanocytic Lesions

Morphological characteristics of actinic keratosis (AK) can be visualized in LC-OCT and this technique can be used for diagnosis and treatment monitoring [29,30,31,32]. Hyperkeratosis, epidermal pleomorphism, and a clearly visible junction were reported as the main changes in AK, correlating well with the histopathology [29,30,31,32]. The proposal of a PRO classification model was based on different stages of the downward extension of basal keratinocytes from the basal layer to the papillary dermis [29]. Grades were classified as PRO I (“crowding” of atypical keratinocytes in the basal layer), PRO II (aggregates of atypical keratinocytes in round nests into the upper papillary dermis), and PRO III (atypical keratinocytes protruding into the dermis), suggesting the chance of non-invasively assessing the progression from AK to invasive squamous cell carcinoma (SCC) [29]. With the attempt to distinguish AK and SCC, two studies compared features characterizing these entities [31,36]. Cellular and architectural changes in the epidermis were found in both AK and SCC (hyperkeratosis/parakeratosis, disruption of stratum corneum, broadened epidermis, basal and suprabasal keratinocyte atypia, and dilated vessels/neoangiogenesis), whereas a well-defined DEJ without broad strands was observed only in AK. A diagnosis of SCC was indicative in the presence of ulceration, acanthosis hampering the visualization of DEJ, and interrupted DEJ [31,36]. Such results suggested that LC-OCT can help in the diagnosis of AK and SCC, as also suggested in another study demonstrating higher diagnostic confidence of this device for keratinocyte skin cancers, compared to RCM [41] (Figure 3). Although LC-OCT can image deeper in the skin, the main technical limitation is the evaluation of hyperkeratotic/acanthotic AK/SCC [29,30,31]. Besides diagnostic application, LC-OCT has been also employed in the treatment monitoring of AK with tirbanibulin, observing that apoptotic keratinocytes (targeted/ring-like cells) were indicative of early-phase therapy response [46] (Table 1).

LC-OCT images of a BCC have clearly correlated with the histopathology since the first introduction paper published in 2018 [16]. Subsequent studies described in detail the morphological criteria of BCC, identifying the dermal lobules as the most useful diagnostic clue [39,61,62]. They are characterized by three components: a grey internal core with a laminated horizontal arrangement (named the “millefeuille pattern”); a middle dark rim surrounding the core; and an external bright rim outlining the lobule [37,39,61,62]. The simultaneous presence of the three structures composing the lobule facilitates the differential diagnosis of BCC from other entities showing dermal lobules, such as sebaceous hyperplasia, dermal nevus, melanoma, cutaneous vascular lesions, and more rare entities [18,23,33,35,42,48,63]. The BCC lobule shape and location within the skin proved to be predictors of a different BCC subtype (superficial, nodular, and infiltrative), with an overall BCC subtype agreement rate of 90.4% with the conventional histopathology [39] (Figure 3). Uncommon subtypes, such as as basosquamous carcinoma and fibroepithelioma of Pinkus, were later investigated by LC-OCT, showing in vivo a clear morphologic correspondence to pathologic features [38,40]. Compared to dermoscopy alone, LC-OCT showed an increase in diagnostic accuracy either for equivocal BCC or including benign and malignant skin lesions [39,41]. In a presurgical setting, LC-OCT was employed to delineate peripheral margins in high-risk BCC, reducing the number of Mohs stages, and for margin adjustment for an infiltrative BCC [53,54]. Its diagnostic ability was also investigated in equivocal skin lesion of the eyelids, obtaining advantages in a sensitive region from a non-invasive technique for more precise treatment planning and an appropriate surgical approach, if required [22,51]. Real-life data supported its important role in the in vivo diagnosis of NMSC, especially BCC, as this imaging technique could spare unnecessary biopsies with a better diagnostic accuracy than dermoscopy and RCM [45,47]. Therapy response was also assessed in superficial BCCs treated with treatment topical cream, otherwise misdiagnosed with dermoscopy [34] (Table 1).

Among non-melanocytic benign lesions, seborrheic keratosis (SK) was described based on a series of histopathologically proven cases. As distinct architectural patterns characterized flat SK, acanthotic SK, hyperkeratotic SK, pigmented reticulated SK, and melanoacanthoma, LC-OCT was deemed to be valuable for the differentiation of SK from clinical imitators [44]. In addition, most of the histological features of clear cell acanthoma and of eccrine poroma have been visualized by LC-OCT, allowing a clear-cut diagnosis [49,50].

Useful diagnostic clues were also given in skin malignancies of rare occurrence, like mycosis fungoides, extramammary Paget’s disease, and Merkel cell carcinoma [42,43,52] (Table 1).

### 3.3. Inflammatory Skin Diseases

A pivotal study investigated 15 adult patients with plaque psoriasis (5 patients), atopic eczema (5 patients), and lichen planus (5 patients) [56]. Psoriasis was characterized by thickening of the stratum corneum and of the epidermis, along with elongated dermal papillae containing dark capillaries [56] (Figure 4). Atopic eczema showed focal areas of thickened and disrupted stratum corneum, irregular epidermis with increased intercellular spaces (spongiosis), and dark circular areas containing scattered keratinocytes (vesicles) [56]. Features of lichen planus were a thickened epidermis (wedge-shaped hypergranulosis), poorly defined DEJ, and dermal papillae with ectatic dermal vessels. Clusters of bright inflammatory cells hampered the visualization of the DEJ [56]. Scalp psoriasis was described in a single case, reporting epidermal hyperplasia with a papillomatous appearance, clusters of bright small cells in the stratum corneum (inflammatory cells), and dark areas filled with amorphous material at the spinous layer (pustules of Kogoj) [57]. Dynamic morphological changes occurring in psoriasis during treatment with biological drugs were evaluated by LC-OCT, with the aid of AI [58]. As related to drug-psoriasis adverse events, this device was used for a diagnosis of a paradoxical skin reaction to ixekizumab that clinically manifested as sebo-psoriasis [59].

LC-OCT appeared very promising as an integrative tool for blistering diseases as the identification of the split level (intraepidermal or subepidermal) was useful in the differential diagnosis of various autoimmune bullous diseases and their mimickers [55] (Figure 4). 

In the spectrum of dermatitis and eczema, consecutive grading reactions to a patch test were defined in vivo: spongiosis and dilatation of blood vessels indicated a weak positive reaction (+). Such features, associated with epidermal thickening, papillomatosis, and a loss in dermal reflectivity, were indicative of a strong positive reaction (++); merging vesicles forming bullae appeared in the case of an extreme positive reaction (+++) [60] (Table 1).

### 3.4. Hair and Nails

LC-OCT can aid in the early diagnosis of inflammatory diseases affecting the adnexa. In a case series, six patients with lichen planopilaris were evaluated with LC-OCT, revealing whitish amorphous material filling the ostium infundibula and covering the hair shaft (infundibular hyperkeratosis), and perifollicular inflammation with bright, ill-defined, coarse collagen fibers [64].

Assessing folliculitis decalvans, the authors reported multiple hair tufts emerging from enlarged hair ostia, bright areas collecting round cells (pustule), and dilated blood vessels in the dermis. Perifollicular inflammatory infiltrate and increased bright, coarse collagen fibers were also detected [65].

Follicular microscopic alterations underlying dermoscopic criteria were investigated in vivo in alopecia areata (AA) (acute and chronic forms) in 65 patients; bright follicular plugs corresponded to yellow dots, and a different caliber of the hair shaft was seen in the presence of broken hairs, black dots, exclamation-mark hairs, and vellus hairs [66]. Later, in patients with AA treated with baricitinib, regrowing vellus hairs were indicative of disease activity and therapy response [67].

LC-OCT findings of tinea capitis were described in a 6-year-old boy. Corkscrew hairs were observed as short, coiled hairs filled with small bright structures suggestive of endothrix-type hyphae hair invasion [68].

LC-OCT was employed for the characterization of healthy nails and various nail disorders [69,70]. Onychomycosis was defined by the presence of bright branching filamentous structures along multiple layers of the nail plate [69,70]. A thickened bright surface with lamellar splitting, wave-like zones of varying intensity in the nail plate were typical findings of nail psoriasis. Lichen planus showed alternating hypo-and hyper-reflective longitudinal bands and crater-like depressions. In subungual melanoma, an irregular nail plate with numerous bright round/ovoidal structures dispersed throughout the entire nail area were observed. Subungual hemorrhage was characterized by a dark area below an irregular bright surface [71] (Table 2).

### 3.5. Infectious Diseases

LC-OCT has been suggested for the diagnosis of some cutaneous infectious and parasitic disorders, as it allows a bedside, quick observation of the etiologic agent. Sarcoptes scabiei, involved in the pathogenesis of scabies, was described as a bright ovoid structure right below the stratum corneum [72,73,80]. The eggs of the mite and its internal organs, as well as its droppings, were clearly visible throughout the burrow, supporting its diagnosis in challenging scenarios [78]. Morphological features of the mite were also clearly described in a case of trombiculosis, appearing with a bright ovoid body with three pairs of legs attached to the stratum corneum [77].

Lacarrubba et al. investigated the LC-OCT features of herpes infection in two patients with herpes simplex and three patients with herpes zoster [74]. In all patients, imaging showed dark intraepidermal roundish areas containing hyperreflective, bright elements, corresponding to vesicular spaces and ballooning, giant multinucleated keratinocytes within the vesicles, as confirmed by Tzanck’s test [74]. The same researcher group investigated molluscum contagiosum infection, observing with LC-OCT the typical architectural changes of this condition: a crateriform invagination composed of large, polygonal, hypo-refractive cells with bright contours [75]. In a case of human cowpox, LC-OCT guided the clinician’s suspicions, thanks to the detection of epidermal disruption and an intraepidermal vesicle with floating cells [76]. To distinguish genital warts from clinical imitators, nine patients with condylomas were investigated by LC-OCT, revealing hyperkeratosis, acanthosis, cells with enlarged nuclei (koilocytes), and dark areas (vessels) in the dermis [81]. Finally, a case of pityriasis rosea was diagnosed with the aid of LC-OCT, showing focal parakeratosis, spongiosis, and erythrocyte extravasation [79] (Table 2).

### 3.6. Pediatric Population

The feasibility of using LC-OCT in pediatric populations was assessed in 67 subjects aged <17 years old. Healthy skin was evaluated in six different stages from infancy to adulthood: infancy (0–12 months, 1–2 years, and 3–4 years), childhood (5–8 years), pre-pubescent period (9–12 years), puberty, and adulthood (13–16 years) in six different body areas, with the aim to assess skin properties in different anatomical areas [84]. The stratum corneum and the mean epidermal width was found to be thickest for the palmar surface of the hand, followed by the back of the hand, the forearm, and the other areas. A correlation between age and growth was found for the different cutaneous localizations [84].

Besides evaluation of the healthy skin, LC-OCT was employed in different skin conditions. Including 73 patients with a final diagnosis of atypical melanocytic lesion (56.7%), spitz nevus (10.9%), vascular lesion (8.1%), congenital nevus (6.8%), halo nevus (5.4%), scabies (4.0%), viral wart (4.0%), pyogenic granuloma (2.8%), and tinea corporis (2.8%), LC-OCT increased diagnostic confidence level with a more accurate management of the skin lesions [86]. 

Another study investigated the differential diagnosis of lichenoid dermatoses of the childhood (i.e., lichen nitidus, lichen spinulosus, lichen striatus, and keratosis pilaris) in 10 patients. The main features of lichen nitidus were roundish dark elements (histiocytes), and ovoid bright elements (lymphocytes) distributed throughout the dermis. Lichen spinulosus, lichen striatus, and keratosis pilaris were instead characterized by a lymphocytic infiltrate, whereas epidermal hyperkeratotic plugs were only found in lichen spinulosus and keratosis pilaris [82]. According to these findings, the authors concluded that LC-OCT may provide useful information for the distinction of lichenoid papular conditions of childhood/adolescence and their clinical simulators [82].

Pyogenic granuloma and agminated spitz nevus were described in single case reports [83,85]. Images of pyogenic granuloma in two children were comparable with the histopathological exam. Indeed, increased inter-keratinocytic spaces (spongiosis) and numerous dark and linear structures (vessels), were clearly seen in both cases [83]. An agminated spitz nevus in a 6-year-old girl showed the key histopathological attributes, bright clusters of melanocytic nests at DEJ, and bright melanophages in the papillary dermis [85] (Table 2). 

### 3.7. Genetic Diseases

Circumscribed palmar hypokeratosis is a rare benign epidermal malformation usually appearing on palms and soles. LC-OCT was performed in three patients, detecting total loss of the stratum corneum with no changes in the stratum spinosum and granulosum, as typically seen in the histopathology [87].

A single case of generalized eruptive histiocytosis (GEH), a rare benign non-Langherans cell histiocytosis, was described in a one-year-old child, developing multiple confluent papules and nodules with a reddish-orange homogenous color along trunk and limbs. LC-OCT revealed bright roundish structures with different sizes distributed in the papillary dermis, corresponding to diffuse xanthomised histiocytes intermingled with eosinophils in histopathology [88].

Multiple skin lesions clinically suggestive of angiokeratomas were studied with LC-OCT in a 15-year-old boy, showing dermal dark areas separated by bright fibrous septa, corresponding to vascular lacunae in the histopathology [89]. The presence of multiple angiokeratomas may represent the first sign of Anderson–Fabry disease in children, when associated with systemic symptoms in various organs, as occurred in this case [89]. 

The autosomal-dominant genodermatoses Darier disease and Hailey–Hailey disease were separately investigated in two papers [90,91]. Eight patients with histopathological diagnosis of Darier’s disease were evaluated with LC-OCT: six patients presented brownish-confluent papules in seborrheic areas, while two patients showed involvement of the intertriginous areas, which appeared partly macerated. LC-OCT showed an epidermis of variable thickness with focal alterations of the stratum corneum, irregular and confluent dark spaces (acantholysis), especially in the supra-basal layers, and sparse, roundish, bright cells with a targetoid appearance (dyskeratotic cells) [90]. Reporting a case of Hailey–Hailey disease with folds involvement, Di Stefani et al. recognized similar LC-OCT features including epidermal hyperkeratosis, focal disruption of the stratum corneum with detaching scales, and supra-basal acantholysis assuming a typical “dilapidated brick wall” appearance [91]. 

A case of Galli–Galli disease presenting with asymptomatic brown papules and lentigo-like macules on the trunk and extremities was diagnosed by combining clinical aspects with imaging. Hence, LC-OCT features of intraepidermal dark areas (acantholysis) with hyper-refractile roundish elements within (dyskeratotic cells), a downward finger-like proliferation of the rete ridges, and basal hyperpigmentation suggested the correct diagnosis, leading to a conservative approach [92] (Table 3).

### 3.8. Skin Vascular Lesions

In a pivotal study, using LC-OCT in a consecutive series of benign and malignant skin vascular lesions, an increased dermal vascularity was found in the different entities [63]. The investigation was based on 71 histopathologically proven skin vascular lesions, including 25 cherry hemangiomas, 15 angiokeratomas, 10 thrombosed hemangiomas, 6 pyogenic granulomas, 5 venous lakes, 4 targetoid hemosiderotic hemangiomas, 4 Kaposi sarcomas, and 2 glomuvenous malformations. A different size/shape for the prominent vascular component, showing in vivo high similarity to that of histopathologic slides, led to supporting an active role of LC-OCT in the differential diagnosis of these entities [63]. Vascular structures were also described in a single case of Kaposi sarcoma of the glans [93] (Table 3).

### 3.9. Cosmetic Applications

Characterization of age-related changes in skin structure and functional properties can now be made thanks to non-invasive imaging techniques, without tissue excision [94]. To quantify superficial dermis thickness and its alterations across aging, LC-OCT has been applied to ex vivo skin biopsies, with a decrease in the dermis thickness noticed in older age groups [95]. The quality of the dermal matrix was scored, evaluating the fibers’ state stratified for different grades, and by training a deep learning model for an automatic assessment. The accuracy was high, and this model showed greater alterations for old and photo-exposed subjects [96]. In another, study quantitative cellular biomarkers of facial skin aging were analyzed using LC-OCT imaging coupled with AI-based quantification algorithms [97]. Such measurements were employed to evaluate the anti-aging effect of dermo-cosmetic agents and procedures targeting the skin at different layers [98,99,100,101,102]. The kinetics of sequential changes with hydradermabrasion were assessed by means of LC-OCT [102]. Images captured after one cycle of treatment showed a decrease in thickness and number of undulations in the stratum corneum, with enhancement in fibrillar collagen in the superficial dermis [99]. When evaluating the effect of a brightening agent on facial pigmented spots, a change in melanin contrast across the epidermis at day 56 was clearly seen, with a decrease in epidermal melanin quantification [101]. In another study, treating post-inflammatory hyperpigmentation with chemical peels resulted in microscopic changes, including an upward migration of melanin in the stratum corneum, without involving the entire epidermis [98].

The use of a diode laser for the removal of the hair was proven to cause in vivo perifollicular inflammation sparing surrounding tissue. Indeed, differences in pre- and post-treatment analysis supported the selective involvement of the melanin pigment in the hair follicles, and the efficacy of a preset device setting in different skin types [100] (Table 3).

### 3.10. Others

Miscellaneous skin conditions of rare occurrence were explored with this new technique. Tognetti et al. reported LC-OCT features of aquagenic keratoderma in two different reports, including overall four patients [103,105]. LC-OCT allowed the capture in vivo of multiple dark spaces in the stratum corneum due to water retention, and dilated ducts of the sweat eccrine glands [103,105]. In a series of pustular skin disorders with a different etiology, applying LC-OCT enabled recognition of a peculiar morphology of the pustules (shape, margins, and cellular content), along with epidermal and adnexal alterations, in different conditions [107]. In a graphite tattoo clinically presenting as a linear blue-grey macule, LC-OCT detected bright particles in the dermis highly indicative of foreign bodies [113]. In addition, in a case of delayed tattoo reaction from red dye, and in five cases of tattoo-induced cutaneous lymphoid hyperplasia, its application was an alternative diagnostic option to incisional biopsy [104,114].

Clinical diagnosis of miliaria crystallina was supported by LC-OCT, revealing its histopathologic substrates like subcorneal vesicles centered by a spiraliform sweat duct [110]. Also in localized mucinosis, acquired digital fibrokeratoma, and lichen planus pigmentosus inversus, LC-OCT correctly revealed the main histopathological findings [106,109,111]. For cutaneous sarcoidosis, the detection of subepidermal granulomas guided the specific site for the biopsy [108]. 

Lastly, exploring the different pathogenetic steps of bullous striae distensae allowed the observation of an increase in fluid leakage from blood vessels, indicating the formation of the vesicula and bulla [112] (Table 3).

## 4. Discussion

In recent decades, there has been a surge in noninvasive diagnostic technologies. However, many tools are still in the research and development phase, and few devices have been widely adopted and used in regular clinical practice, viz., RCM and OCT [1,2,3]. These techniques rely on medical devices that offer different analysis levels in vivo, and their use increases the accuracy of diagnosis and allows unnecessary biopsies to be avoided [1,2,3]. Indeed, the high resolution and depth capacities of RCM and OCT, respectively, enable the dermoscopic diagnosis to be refined, with the transversal visual giving similar information to histopathologic sections [13,14]. 

The combination of both RCM and OCT in one medical device appears to be the technique of choice for diagnosing skin lesions in real-time. Indeed, merging the high-resolution power of RCM with the vertical imaging and deep penetration capabilities of OCT, LC-OCT reduces the technical limits of these devices, obtaining real-time three-dimensional (3D) multi-planar imaging (vertical and horizontal) of the epidermis and superficial dermis at the cellular level, reproducing a “virtual biopsy” [16,17]. As it is considered a bridge between dermoscopic morphology and histopathology, LC-OCT has aroused a great interest in the scientific community for its capacity to increase the diagnostic accuracy in keratinocyte skin cancers (when in the hands of expert clinicians), with evolving applications in different fields of dermatology, mainly relating to inflammatory conditions and infectious diseases. However, it must be taken into account that the deeper part of invasive tumors and dermal features of inflammatory diseases may be missed as its penetration depth reaches 500 μm. 

Because of its novelty, the current literature is limited and mainly relies on case reports/series and few original research studies. This represents one of the main limitations of this systematic review and explains why meta-analyses could not be conducted, as the studies addressed somewhat different clinical questions and were methodologically heterogeneous. Case reports and case series were, however, included as they may provide important details that are relevant and can contribute to a change in clinical practice. The restriction to English-language studies may have represented a methodological bias.

An important challenge of LC-OCT is represented by the high cost of the device, besides the achievement of an adequate expertise in image reading, which limits its use to large academic and research centers. In recent decades, the benefit of the systematic use of RCM was proven, as it significantly affected the number of benign lesions excised, resulting in a significant cost–benefit advantage. In a single-center study, 4320 unnecessary excisions were avoided in a year, saving over EUR 280,000 thanks to its use [115]. In another investigation, 50.2% of biopsies were avoided by identifying benign lesions with RCM [116]. It is premature to undertake cost-effectiveness analysis derived from the systematic application of LC-OCT because of its recent introduction and scarce diffusion.

## 5. Conclusions

Significant progress in imaging systems has led technology to become a ubiquitous part of modern clinical diagnosis. Even though increasing data and large-scale studies are warranted to support and confirm the existing evidence, LC-OCT seems to be an adjunctive in vivo tool that provides key clues in a diagnostic setting, as assessed in the current literature.

## Figures and Tables

**Figure 1 diagnostics-14-01821-f001:**
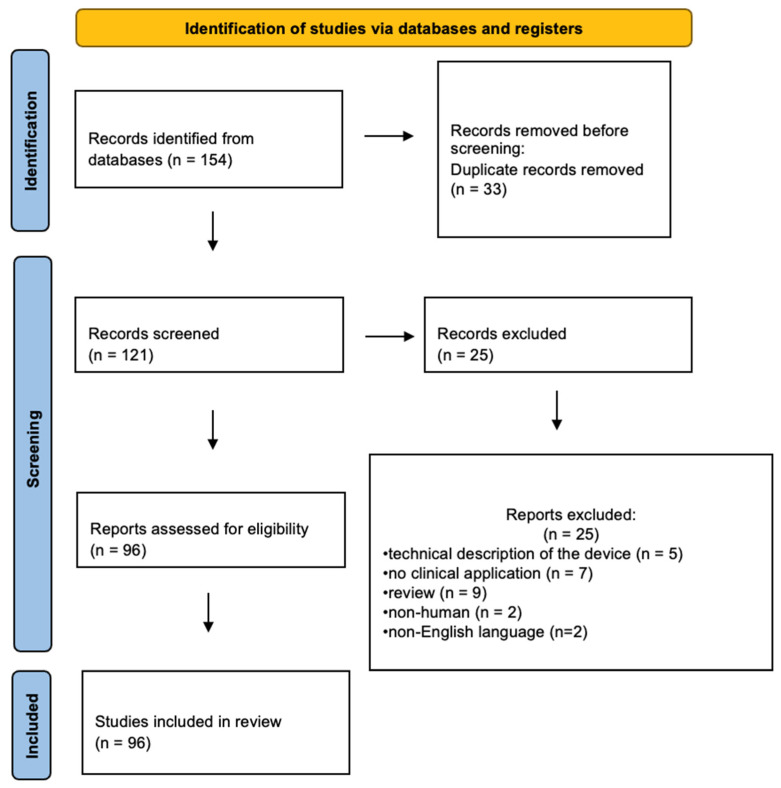
PRISMA flowchart diagram (adapted from Page et al., 2021) [20].

**Figure 2 diagnostics-14-01821-f002:**
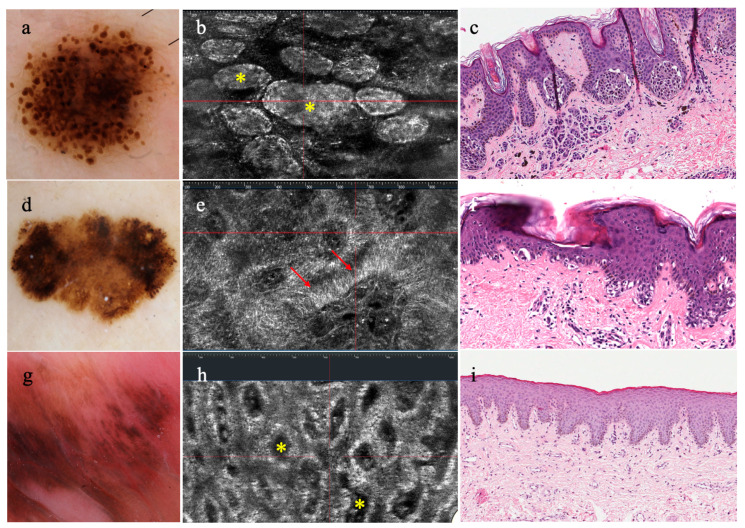
**Dermatoscopy, horizontal LC-OCT, and histopathology of compound nevus, melanoma, genital melanosis:** (**a**) compound nevus with globular pattern (**b**) associated with junctional nests (yellow asterisks) in LC-OCT and (**c**) histopathology; (**d**) melanoma showing atypical network and eccentric hyperpigmented areas in dermatoscopy, (**e**) due to atypical junctional melanocytes (red arrows) in LC-OCT (**f**) clearly seen in histopathology; (**g**) melanosis with dermatoscopic features of parallel pattern (**h**) that reveals a draped pattern (yellow asterisks) in LC-OCT, (**i**) corresponding to hyperpigmentation of the basal keratinocytes in histopathology.

**Figure 3 diagnostics-14-01821-f003:**
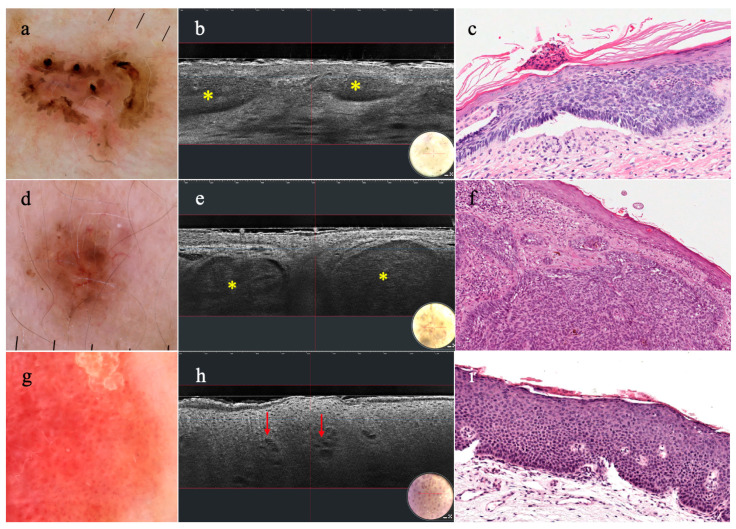
**Dermatoscopy, vertical LC-OCT, and histopathology of superficial basal cell carcinoma (sBCC), nodular basal cell carcinoma (nBCC), and Bowen disease:** (**a**) sBCC showing leaf-like areas and erosions in dermatoscopy, (**b**) identified under LC-OCT as hemispheric lobules attached to the epidermis (yellow asterisks), (**c**) relating to tumor islands in histopathology; (**d**) nBCC with arborizing micro vessels and brown structureless areas in dermatoscopy, (**e**) visualized as ovoid-shaped lobules separated from the epidermis (yellow asterisks) in LC-OCT, and (**f**) histopathology; (**g**) Bowen disease with dermatoscopic pattern of glomerular vessels and a scaly surface, (**h**) revealing hyperkeratosis and dilated vessels (red arrows) in LC-OCT, and (**i**) histopathology.

**Figure 4 diagnostics-14-01821-f004:**
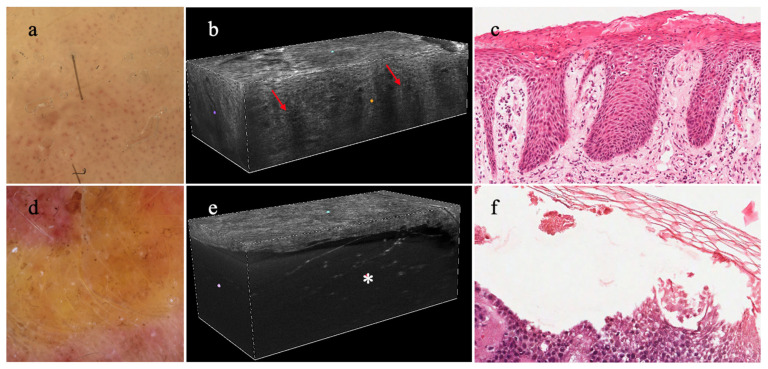
**Dermatoscopy, 3D LC-OCT, and histopathology of plaque psoriasis and bullous pemphigoid:** (**a**) plaque psoriasis with a regular distribution of red dots and white scales in dermatoscopy, (**b**) showing elongation of the dermal papillae (red arrows) and dilated vessels in LC-OCT (**c**) clearly observed in histopathology; (**d**) bullous pemphigoid characterized by a yellow translucent structureless area in dermatoscopy, (**e**) justified by a junctional split creating a bulla with floating elements in LC-OCT (white asterisk), (**f**) as seen in histopathology.

**Table 1 diagnostics-14-01821-t001:** Melanocytic and non-melanocytic skin lesions, and inflammatory and autoimmune diseases.

	Authors and Year of Publication	Disease	Number of Patients
melanocytic lesions	Lenoir et al., 2021 [21]	dermal nevi	7
	Schuh et al., 2022 [23]	nevi and melanoma	75
	Perez-Anker et al., 2022 [26]	morphological evaluation of melanocytic lesions	12
	Verzi’ et al., 2023 [24]	lentigo maligna	1
	Zeinaty et al., 2023 [28]	pigmented lesions of genital area	9
	Wolswijk et al., 2023 [25]	nodular melanoma	1
	Soglia et al., 2024 [27]	melanocytic skin tumours	
non- melanocytic lesions	Dubois et al., 2018 [16]	BCC	86
	Ruini et al., 2021 [29]	AK	50
	Ruini et al., 2021 [30]	BCC	52
	Ruini et al., 2021 [31]	keratinocyte skin cancers	73
	Lenoir et al., 2021 [32]	AK	16
	Lenoir et al., 2021 [33]	sebaceous hyperplasia	12
	Verzì et al., 2021 [34]	monitoring of superficial BCC treated with imiquimod 5% cream	12
	Lacarrubba et al., 2021 [35]	Xanthogranuloma	1
	Cinotti et al., 2021 [36]	AK and SCC	158
	Cappilli et al., 2021 [37]	BCC	1
	Cappilli et al., 2022 [38]	fibroepithelioma of Pinkus	5
	Gust et al., 2022 [39]	BCC	154
	Cappilli et al., 2022 [40]	basosquamous carcinoma	8
	Cinotti et al., 2022 [41]	keratinocyte skin cancers	52
	Di Stefani et al., 2023 [42]	extramammary Paget disease	1
	Di Stefani et al., 2023 [22]	eyelid skin lesions	51
	Soglia et al., 2023 [43]	merkel cell carcinoma	1
	Lenoir et al., 2023 [44]	seborrheic keratosis	29
	Donelli et al., 2023 [45]	keratinocyte skin cancers	360
	Lacarrubba et al., 2023 [46]	treatment monitoring of AK with tirbanibulin	10
	Cinotti et al., 2023 [47]	BCC	62
	Aktas et al., 2023 [48]	differential diagnosis of infiltrative BCC with scar-like lesion	4
	Cortonesi et al., 2023 [49]	clear cell acanthoma	7
	Maione et al., 2023 [50]	eccrine poroma	1
	Verzì et al., 2023 [51]	BCC, SCC, nevi, seborrheic keratoses, pyogenic granuloma, trichilemmal cysts, hidrocystoma	28
	Soglia et al., 2023 [52]	cutaneous mycosis fungoides	10
	Jacobsen et al., 2024 [53]	surgical planning for recurrent infiltrative BCC	
	Paradisi et al., 2024 [54]	preoperative evaluation of high-risk BCC	60
inflammatory and autoimmune skin diseases	Tognetti et al., 2021 [55]	autoimmune bullous diseases	30
	Verzì et al., 2022 [56]	psoriasis, eczema and lichen planus	15
	Truong et al. 2023 [57]	scalp psoriasis	1
	Orsini et al., 2024 [58]	psoriasis treatment monitoring	17
	Falcinelli et al., 2024 [59]	sebo-psoriasis-like dermatosis reaction to ixekizumab	1
	Russo et al., 2024 [60]	evaluation of Patch Test	

BCC = basal cell carcinoma; AK = actinic keratosis; SCC = squamous cell carcinoma.

**Table 2 diagnostics-14-01821-t002:** Hair and nails, skin infectious diseases, and pediatric skin conditions.

	Authors and Year of Publication	Disease	Number of Patients
hair and nails	Hobelsberger S et al., 2023 [70]	healthy nails and onychomycosis	13
	Kurzeja et al., 2023 [64]	lichen planopilaris	6
	Eijkenboom et al., 2024 [71]	leukonychia, subungual haemorrhage, psoriasis, lichen planus, longitudinal melanonychia, subungual melanoma and onychomycosis	16
	Eijkenboom et al., 2024 [69]	onychomycosis	100
	Lacarrubba et al., 2024 [66]	Alopecia areata	65
	Verzì et al., 2024 [67]	Alopecia areata	10
	Kurzeja et al., 2024 [65]	folliculitis decalvans	5
	Falcinelli et al., 2024 [68]	tinea capitis	1
skin infectious disorders	Ruini et al., 2020 [72]	scabies	1
	Ruini et al., 2021 [73]	scabies	1
	Lacarrubba et al., 2021 [74]	herpes infection	5
	Verzì et al., 2021 [75]	molluscum contagiosum	6
	Cortonesi et al., 2022 [76]	cowpox virus skin infection	1
	Cappilli et al., 2022 [77]	trombiculosis	1
	Orsini et al., 2023 [78]	nodular scabies	1
	Pathak et al., 2023 [79]	pityriasis rosea	1
	Idoudi et al., 2024 [80]	scabies	1
	Cinotti et al., 2024 [81]	genital warts, molluscum contagiosus, Fordyce’s spot, acquired lymphangiomas	14
pediatric conditions	Tognetti et al., 2021 [82]	lichenoid dermatoses	10
	C Gallay et al., 2022 [83]	pyogenic granuloma	2
	Del Río-Sancho, 2023 [84]	healthy skin	67
	Del Río-Sancho et al., 2023 [85]	agminated spitz nevus	1
	Cappilli S. et al., 2023 [86]	melanocytic lesions, vascular lesions, scabies, viral warts, pyogenic granulomas, tinea corporis	73

**Table 3 diagnostics-14-01821-t003:** Skin vascular lesions, genetic diseases, cosmetic applications, and others.

	Authors and Year of Publication	Disease	Number of Patients
skin vascular lesions	Tognetti et al., 2021 [93]	kaposi sarcoma	1
	Cappilli et al., 2023 [63]	cherry haemangiomas, angiokeratomas, pyogenic granulomas, venous lakes, targetoid haemosiderotic haemangiomas, kaposi sarcomas, extraungual glomus tumours	50
genetic diseases	Tognetti et al., 2020 [87]	circumscribed palmar hypokeratosis	3
	Bruzziches et al., 2022 [88]	generalized eruptive histiocytosis	1
	Tognetti et al., 2022 [89]	Anderson–Fabry disease	1
	Di Stefani et al., 2023 [91]	Hailey–Haiey disease	1
	Verzì et al., 2023 [90]	Darier’s disease	8
	Maione et al., 2024 [92]	Galli–Galli disease	1
cosmetic applications	Chauvel-Picard et al., 2022 [94]	in vivo quantification of healthy epidermis	8
	Pedrazzani et al., 2020 [95]	quantification of superficial dermis thickness	36
	Breugnot et al., 2023 [96]	dermal matrix quality assessment	57
	Bonnier et al., 2023 [97]	quantitative biomarkers of facial skin aging	100
	Razi et al., 2022 [98]	effects of a chemical peel on post-inflammatory hyperpigmentation	1
	Razi et al., 2024 [99]	early effects of diamond-tipped microdermabrasion	8
	Razi et al., 2024 [100]	cutaneous effects of diode laser	3
	Jdid et al., 2024 [101]	skin dark spot mapping and evaluation of brightening product efficacy	26
	Razi et al., 2024 [102]	hydradermabrasion	8
Others	Tognetti et al., 2020 [103]	aquagenic keratoderma	3
	Tognetti et al., 2020 [104]	delayed Tattoo Reaction	1
	Tognetti et al., 2021 [105]	aquagenic keratoderma	1
	Tognetti et al., 2022 [106]	acquired digital fibrokeratoma	1
	Tognetti et al., 2022 [107]	pustular skin disorders	19
	Thamm et al., 2023 [108]	cutaneous sarcoidosis	1
	Palmisano et al., 2024 [109]	solitary cutaneous focal mucinosis	1
	Verzì et al., 2024 [110]	miliaria crystallina	1
	Lamberti et al., 2024 [111]	lichen planus pigmentosus inversus	1
	Falcinelli et al., 2024 [112]	bullous striae distensae	1
	Castellano et al., 2024 [113]	graphite tattoo	1
	Ariasi et al., 2024 [114]	tattoo-associated cutaneous lymphoid hyperplasia	5

## Data Availability

The data presented in this study are available on request from the corresponding author.

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
