# Peer review of "Line-Field Confocal Optical Coherence Tomography: A New Skin Imaging Technique Reproducing a “Virtual Biopsy” with Evolving Clinical Applications in Dermatology"

_diagnostics, 2024, doi:10.3390/diagnostics14161821_

Round 1

Reviewer 1 Report

Comments and Suggestions for Authors

The authors review several research articles related to Line-field confocal optical coherence tomography (LC-OCT) to conduct "virtual biopsies" of the skin. A review of its application across various skin conditions, including cancers, inflammatory and infectious diseases, genetic disorders, cosmetic procedures, and other uncommon conditions was investigated.

The manuscript reviews these papers in a systematic way. However, there are some issues to be addressed. Hence, I recommend publication of the manuscript only after significant improvements are made.

To begin with, I advise the authors to thoroughly review the manuscript, addressing any typos as well as contextual and structural errors. I also recommend including a subsection that briefly explains the state-of-the-art LC-OCT. This should feature schematics of the system and a short text explaining the optical path. Additionally, I suggest the authors make the following corrections:

1.     I recommend the authors to avoid using any abbreviations before they are defined; some examples include LC-OCT and PRISMA in the abstract sections, as well as SCC (line 200), and AK (line 191) are used before they are defined. Some are defined twice, such as RCM at line 54 and 79. Also at line 35, the phrase; ‘…able to o reproduce…’ should be corrected.

2.     In Introduction section line 92, the authors ignored the high-resolution capability of latest OCT technologies that provide around 2µm theoretical resolution.

3.     I suggest the authors correct numbering, capitalization and font of subsection. Line 106, 261, 190, 300, 329…subsection titles capitalization and font should be consistent in different subsections. Some are all capital letters and others are not. I suggest avoiding capitalization of subsections.  Subsections in section 2 should be numbered, like 2.1 Search Methods…, 2.2. Data collection… This should be consistent in all other section subsections. Subsubsection at line 108 and 115 are hard to recognize. They should at least be numbered or bold and be separated from the text using : instead of a full stop (.).

4.     Unnecessary and extra spacing between titles, paragraphs, and figure and text should be avoided. Examples include line 47, 50, 105, 107, 286, 288 and others.

5.     Titles of Subsections such as at line 157, 261, 329 shouldn’t be at the end of the page.

6.     Figure caption should be consistent. I suggest the authors go through and correct them. Figure 1 caption should be at the bottom of the figure, Figure 2 caption should be all in the same page as the figure. In all figure captions, the authors should mention where they take it from. The phrase should be something like: ‘taken from [citation]’.

7.     Text in conclusion section has different font size compared to other sections. This should be corrected.

8.     Please check consistency in the reference section, such as journal naming and abbreviations.

Comments on the Quality of English Language

Dear Authors 

Please go through the manuscript to check and correct any typos and structural errors.

best,

Reviewer 2

Reviewer 2 Report

Comments and Suggestions for Authors

General Comments: The manuscript deals with a new technique in the field of noninvasive diagnostics of skin: "Line-field confocal optical coherence tomography: a new skin imaging technique reproducing a 'virtual biopsy' with evolving clinical applications in Dermatology." The authors prepare a comprehensive review of applications of line-field confocal optical coherence tomography in dermatology with coverage of different skin diseases. The topic is current and quite relevant to the medical community in augmenting the efficiency of diagnosis with minimal discomfort to the patient. However, there are a few crucial areas that need enhancement to strengthen the manuscript.

Introduction: The introduction describes with utmost clarity the need for advanced diagnostic tools in dermatology and its quoted the limitations involved with biopsy techniques and benefits of the non-invasive imaging technologies; however, a more detailed discussion on the specific benefits of LC-OCT over its competitors and over the presently used methods such as RCM and OCT is desired. What are the specific technical features of LC-OCT that make it a superior diagnostic tool over them? Moreover, the introduction could be more developed and current with the research situation by including some of the latest studies and technological advances in the related field.

Materials and Methods: It's a good thing that the authors are following the PRISMA guidelines for the systematic review. The criteria for selecting studies to be included in the review need to be more transparent. How were discrepancies between reviewers reconciled, and what was used as a selection criterion for studies where reviewers disagreed? In addition, excluding studies in other languages but English could lead to a bias. Have the authors considered the potential influence of excluding significant research that is published in other languages? It would also be helpful to present a very brief discussion about the limitations of the search strategy and about any possible biases.

Results: This section is very detailed about what has been found regarding the different skin conditions. This section could be improved with focused synthesis of the findings and displaying the data through sub-sections or tables to read and compare the information better. Again, while the authors indicated the number of studies they were able to identify and review, they did not critically appraise the quality of the studies. What were the main limitations or strengths of the included studies? Were there any significant methodological flaws or biases that might affect the overall conclusions? Finally, take into account a meta-analysis as a way to quantitatively evaluate the accuracy of LC-OCT for diagnosis if this is possible. Explain why if it not.

Discussion: The discussion identifies well the potential of LC-OCT for multiple dermatologic applications and as a bridging technology between dermoscopy and histopathology. But it is far too optimistic because it does not give enough focus to the limitations and difficulties with the technique. What are the major hurdles for wide clinical adoption of LC-OCT? Are there any technical limitations or cost considerations that should be addressed? Moreover, the discussion may be further elaborated with comparisons made to other emerging imaging technologies and a critical assessment of where LC-OCT stands in the broader context of dermatological diagnostics.

Conclusion: In conclusion, although the manuscript described a comprehensive review of LC-OCT and its applications in dermatology, the need for critical and detailed analyses of the current research landscape is much called for. Addressing the questions and concerns raised in this review will significantly enhance the quality and impact of the manuscript.

Comments on the Quality of English Language

Minor editing is required

Round 2

Reviewer 1 Report

Comments and Suggestions for Authors

Dear Authors

I sincerely appreciate the time and effort you have dedicated to carefully reviewing and considering my comments. Your thoughtful attention to my suggestions and the subsequent improvements made to the manuscripts reflect a strong commitment to producing high-quality work. Now, I believe that the final versions of the manuscript is as robust and polished as possible.

Minor correction: Could you check subsubsections at line 109 and 116. I still believe that they should be separated from the text by a ':'

Best regards,

Reviewer 1

Reviewer 2 Report

Comments and Suggestions for Authors

Improved, thank you for responding to my queries